Taxon-dependent diversity response along a temperate elevation gradient covered by grassland

Santoandré Santiago santoandre@ege.fcen.uba.ar 1
Ramos Carolina Samanta 1
Picca Pablo 2
Filloy Julieta 1
1 Departamento de Ecología, Genética y Evolución, Facultad de Ciencias Exactas y Naturales (IEGEBA-CONICET), Universidad de Buenos Aires , Ciudad de Buenos Aires , Argentina
2 Departamento de Biodiversidad y Biología Experimental, Facultad de Ciencias Exactas y Naturales, Universidad de Buenos Aires , Ciudad de Buenos Aires , Argentina
Silva Daniel
Electronic publication date: 2024 Jun 21
Publication date: 2024
Volume: 12
Electronic Location ID: e17375
Received 2023 Aug 16; Accepted 2024 Apr 19
Copyright: ©2024 Santoandré et al.
Copyright year: 2024
Copyright holder: Santoandré et al.
License: This is an open access article distributed under the terms of the Creative Commons Attribution License, which permits unrestricted use, distribution, reproduction and adaptation in any medium and for any purpose provided that it is properly attributed. For attribution, the original author(s), title, publication source (PeerJ) and either DOI or URL of the article must be cited.
License URL: https://creativecommons.org/licenses/by/4.0/

Keywords: Energy hypothesis, Mountain, Multi-taxa, Nestedness, Turnover, Ants, Spiders, Birds, Plants, Climatic driver

Funding: Buenos Aires University and Consejo Nacional de Investigaciones Científicas y Técnicas (CONICET) The work was funded by the Buenos Aires University and Consejo Nacional de Investigaciones Científicas y Técnicas (CONICET). The funders had no role in study design, data collection and analysis, decision to publish, or preparation of the manuscript.

==============================
Elevational gradients constitute excellent systems for understanding the mechanisms that generate and maintain global biodiversity patterns. Climatic gradients associated with elevation show strong influence on species distribution in mountains. The study of mountains covered by the same habitat type is an ideal scenario to compare alternatives to the energy hypotheses. Our aim was to investigate how changes in climatic conditions along the elevational gradient drive α- and β-diversity of four taxa in a mountain system located within a grassland biome. We sampled ants, spiders, birds and plants, and measured climatic variables at six elevational bands (with 10 sampling sites each) established between 470 and 1,000 masl on a mountain from the Ventania Mountain System, Argentina. Species richness per site and β-diversity (turnover and nestedness) between the lowest band and upper sites were estimated. For most taxa, species richness declined at high elevations and energy, through temperature, was the major driver of species richness for ants, plants and birds, prevailing over productivity and water availability. The major β-diversity component was turnover for plants, spiders and birds, and nestedness for ants. The unique environmental conditions of the upper bands could favour the occurrence of specialist and endemic species.

Introduction

Life is heterogeneously distributed on Earth and the attributes of biological communities most often vary in space, giving rise to patterns. The study of the spatial or geographical processes underlying these patterns is a topical issue in ecology (Hawkins et al., 2003). In this regard, elevational gradients are considered valuable systems to better understand the mechanisms involved in their generation and maintenance (Lomolino, 2001; McCain, 2009; Peters et al., 2016; Ramos et al., 2021). Mountain gradients offer a variety of conditions useful to answer particular questions about drivers of diversity (McCain, 2009; Sundqvist, Sanders & Wardle, 2013). However, it is still unclear how climatic drivers affect diversity along elevational gradients within a single biome (e.g., grassland) because most of the studies were carried out on elevation gradients with a sequence of habitat types (e.g., steppe, forest and grassland, as Werenkraut, Fergnani & Ruggiero, 2015).

Species richness has been the most studied component of diversity along elevational gradients (McCain & Grytnes, 2010) and energy and water are known to be the main drivers of the geographical distribution and diversity patterns of species (Hawkins et al., 2003). An increase in incident energy favours an increase in taxonomic richness by providing support to species that otherwise would be limited by physiological constraints. On this basis, species richness in mountains is expected to decrease with decreasing temperature (Sanders et al., 2007). In animal communities, energy and water conditions may limit species richness by affecting the availability of trophic resources given by the productivity of the systems (Mittelbach et al., 2001). Temperature decreases monotonically with elevation, while water availability may decrease or increase and their combined effect results in different patterns of primary productivity (McCain, 2009). Therefore, peaks in species diversity at mid-elevations were expected on mountains characterized by dry and arid bases, while a reduction in species diversity was expected on mountains with moist and warm bases (McCain, 2007).

Changes in the species composition of communities (β-diversity) are crucial for understanding the causes of variation in spatial patterns of species distribution (Liira et al., 2008). It has been found that changes in the taxonomic composition of species may be associated with changes in environmental conditions or geographical distance (MacArthur, 1972; Hubbell, 2001). The connection with shifts in environmental factors offers evidence that variations in species composition stem from varying degrees of species specialization across different dimensions of their ecological niche (MacArthur, 1972). Conversely, the correlation with geographical distance points towards limitations in species dispersal (Condit et al., 2002). Therefore, the degree of taxonomic similarity between communities separated by a short geographical distance and in the absence of geographical barriers to dispersal will depend on the similarity of their environments and the species capacity to survive in the environment (Qian, Ricklefs & White, 2005; Steinitz et al., 2006).

Mountains are characterized by a high degree of environmental variation over a short geographical distance, making them an ideal setting to investigate changes in the dissimilarity of species composition resulting from environmental changes (Jankowski et al., 2009; Tang et al., 2012). However, the relative contribution of the components of β-diversity (i.e., turnover and nestedness) to these changes remains unclear. In this regard, mountains with homogeneous habitat types provide an opportunity to explore changes in species composition due to other factors. A nested pattern is expected when a subset of generalist species found at lower elevations can survive the environmental conditions at upper elevations, while a turnover pattern is likely to occur if species are specialists that only persist at higher elevations. Therefore, the analysis of the patterns of β-diversity components among assemblages allows us to delve deeper into the causes underlying changes in α-diversity along elevational gradients.

A multi-taxa approach is a promising method to elucidate complex community diversity patterns along elevational gradients (Peters et al., 2016; Di Nuzzo et al., 2021). Since assemblages composed of the same taxon tend to share many ecological niche characteristics, they are expected to respond similarly to environmental changes, giving rise to idiosyncratic responses. The multi-taxa approach involving phylogenetically distant taxa allows us to reduce this effect. Indeed, broadening the taxonomic scope may lead to changes in some diversity patterns and reveal the importance of potential climate and environmental drivers of diversity (Di Nuzzo et al., 2021). Our aim was to analyze how environmental changes, along an elevational gradient within a grassland biome, modulated the richness and composition of species assemblies of different taxa (birds, ants, spiders, and plants). We expect that species richness will decline as available energy decreases, and beta diversity components will show a nestedness pattern due to the loss of generalist species, or turnover due to the presence of specialists in certain areas of the elevation gradient.

Methods

Study area

This study was conducted in the Ernesto Tornquist Provincial Park (38°03′41″S, 61°59′18″W) located in the centre of the Ventania System, which is one of the few conservation areas in the Pampean grasslands (Permission number: 057/11). Ventana Mountain ranges from 450 to 1,130 masl (Ponce, 1986). The climate is temperate, with a mean annual temperature of 15 °C and a mean annual rainfall of 700 mm (Kristensen & Frangi, 1995).

The Ventania Mountain range is located southwest of Buenos Aires province, Argentina, in the Pampas Plain. It is an isolated system characterized by steep slopes and a rugged terrain with numerous ravines, and the landscape is dominated by grasslands without trees or shrubs. This mountain system is ancient and dates back to 280–500 million years ago; it supports a rich biodiversity and a high level of endemism for several taxa (Kristensen & Frangi, 1995). Some examples of endemic vegetation are Senecio ventanensis (Asteraceae), Poa iridifolia (Poaceae), Adiantum thalictroides (Adiantaceae) and Olsynium junceum (Iridaceae) (Cuevas & Zalba, 2009). Moreover, a sun-spider species collected in the present study was described as a new species, Gaucha casuhati (Botero-Trujillo, Ott & Carvalho, 2017).

Study Design

A series of six elevational bands were established encompassing most of the elevational gradient of Ventana Mountain, from the valley at the foot of the mountain to near the top, separated between 47 to 108 m of elevation (i.e., at average heights of 471, 517, 625, 723, 844 and 1001 masl). The sampling points in the higher band were arranged over a higher altitude range, from 920 to 1,095 masl, due to the lack of space to separate sampling points. We have sampled in homogeneous areas within each elevational band (i.e., avoiding stream ravines). Fieldwork was carried out in spring and summer (mid-November 2011 to mid-January 2012).

We used phylogenetically distant taxa: ants, spiders, birds, and vascular plants. These groups were selected because they have proven to be sensitive to environmental changes along elevational gradients, are very abundant and diverse, and provide important ecosystem functions (Sekercioglu, 2012; Peters et al., 2016). Ants tend to exhibit high local diversity, performing numerous ecological functions in a wide variety of niches (Hölldobler & Wilson, 1990). They are thermophiles and their diversity tends to decrease at low temperatures (Hölldobler & Wilson, 1990; Sanders et al., 2007). Spiders are a ubiquitous component of invertebrate assemblages and important generalist predators in ecosystems (Wise, 1995). Their richness is associated with prey availability, which is often highly correlated with habitat structural complexity and primary productivity (Aisen et al., 2017). Birds are sensitive to alterations in vegetation structure (Phifer et al., 2017) and respond to changes in primary productivity along gradients (Harrower et al., 2017). Plants are sensitive to changes in abiotic conditions and provide habitat for animals, influencing their diversity and distribution (Nic Lughadha et al., 2005). Water and energy are crucial for plant physiological processes, thus having direct effects on plant diversity (O’Brien, 2006).

Sampling of ants and spiders and taxonomic identification

At each elevational band, we established ten sampling points spaced at least 50 m apart and installed a pitfall trap at each sampling point, making up a total of 60 traps. Pitfall traps are widely used to obtain representative samples of ant and spider assemblages (Agosti et al., 2000; Pinto et al., 2018). These consisted of plastic containers (500 ml in volume and 85 mm in diameter) partially filled with a solution of 150 ml of propylene glycol: water (1:2). Traps were inserted flush with the ground surface and protected by a plastic cover to avoid flooding by rain. They remained open for three one-week periods. Captured specimens were identified to species level whenever possible or to morphospecies (hereafter referred to as species) based on diagnostic characters of the genera or families, using taxonomic keys and specialist consultations.

Bird survey

Birds were surveyed using the point-count method with a fixed 50-m radius (Ralph et al., 1996). At each elevational band, ten point-count sites were established 300 m apart from each other, except for the highest band where the separation was 150 m because it was not possible to space the sites further since there was no more area available at that elevation. Surveys were conducted on clear and calm days, from dawn to the following 4 h. At each point-count site, we recorded all birds seen or heard over 5 min, except for birds flying overhead. The same two observers conducted all surveys.

Plant sampling and identification

Vascular plant species were estimated using 1-m2 and 16-m2 quadrats for herbaceous plants and shrubs, respectively (Kent, 2012). This procedure was repeated twice near the point-count sites for birds (thus resulting in ten points per elevational band), and the species recorded in each subsample were pooled to obtain a species list by site. Bryophytes were not studied. Plant species that could not be identified in the field were dried for further identification in the laboratory. To represent the changes in endemic species with elevation, we selected the endemic plants of the southern sierras of Buenos Aires province from our collected species. Plant taxonomy and distribution follow the Catálogo de Plantas Vasculares del Cono Sur and the online update (http://www.darwin.edu.ar).

Environmental gradient characterization

At each elevational band, an automatic sensor of temperature (HOBO U23002) was placed 10 cm above the ground level, in the centre of each band, and gathered data at 6-h intervals for one month in December (Werenkraut, Fergnani & Ruggiero, 2015). The average temperature per band was calculated including all records for each elevation band. Primary productivity was estimated at each level using a soil-adjusted vegetation index (SAVI) (Huete, 1988). SAVI maps covered the entire length of the elevational gradient and the average value for each elevational level was extracted. The maps were derived from Landsat-5 satellite images of 900 m2 resolution, taken in the springs of 2010 and 2011. To estimate water availability, constant volume soil samples were taken at ten points in each elevation band. The samples were weighed in the field and after being dried in an oven at 70 °C for one week. The soil water content of each sample was calculated as the difference between wet and dry weight. The average water content was estimated for each elevation band. Mean annual temperature (°C, BIO1) and annual precipitation (mm, BIO12) data were obtained from WorldClim V.II (Fick & Hijmans, 2017), already used in mountains (Ramos et al., 2021). Thus, we obtained a value of each variable for each sampling point and then the average per band was calculated.

Data analyses

The description of environmental changes along the elevation gradient was carried out by performing general linear models (GLM), with each environmental variable as the dependent variable and elevation as the explanatory variable, using glm function of stats package. All analyses were carried out with R software project (R Core Team, 2022). Species richness was calculated as the number of species of each taxon per sampling site. The elevational trend of species richness was analyzed using generalized linear models (GLMs) with this variable as a function of elevation. The distributions used were Poisson for, ants, spiders, birds and endemic plants richness and negative binomial for plants to avoid overdispersion with stats, glmmTMB (Magnusson et al., 2017) and visreg packages (Breheny & Burchett, 2017). The significance of each model was obtained by comparing with its respective null model using the anova function. Sample coverage by elevation band, for each taxon separately, was estimated with iNEXT package (Hsieh, Ma & Chao, 2016). We used a multi-model approach to identify the main environmental factors driving changes in species richness along the elevational gradient. For each taxon, we ran GLMs with species richness as the response variable and environmental variables as the independent ones. The error distributions used were Poisson for, ants and birds richness and negative binomial for plants to avoid overdispersion with stats and glmmTMB packages (Magnusson et al., 2017). Pearson correlation between environmental variables was explored using the cor function of stats package. To avoid collinearity between explanatory variables for each model, we evaluated the variance inflation factor (VIF) in the over-parameterised model, discarding the variables with the highest VIF consecutively until none had a value greater than five (Dormann et al., 2013). Then, the effect of the explanatory variables was studied using the anova function of stats package. The main explanatory variable was determined as the lowest p-value less than 0.05.

Changes in species composition along the elevational gradient were modelled as taxonomic dissimilarity considering the turnover and nestedness components (Baselga, 2010). To represent the community that inhabits the vast lowlands all species recorded at the sampling sites of the lowest elevational band were pooled. Each taxon was grouped separately to obtain the different reference communities. To investigate variations in the mountain community, the change in species composition of the assemblages along the elevational gradient was estimated by comparing the dissimilarity at the lowest band with that recorded at each sampling site of the upper bands, based on species presence data (Santoandré et al., 2019; Fontana et al., 2020). Sorensen dissimilarity and its components were estimated using beta.pair function of package betapart (Baselga et al., 2013). Finally, we compared patterns of taxonomic dissimilarity between taxa along the elevation gradient using a GLM with beta distribution, since the dissimilarity values fell within the (0–1) interval (Ramos et al., 2018). The significance of each model was obtained by comparing with its respective null model using the anova function. We only compared the trends of the β-diversity components because the magnitude of the effect was influenced by data pooling. Null models of community assembly were made to investigate the processes that are driving observed β-diversity patterns (Bishop et al., 2015; Foord & Dippenaar-Schoeman, 2016). Null models were performed generating 999 random assemblages using the independent swap method of the randomizeMatrix function of picante package (Kembel et al., 2010). This method maintains sample species richness while shuffling species co-occurrence across sites. To analyse the relationship between the observed β-diversity and that expected by chance, the standardized effect sizes (SES) were calculated (Swenson, 2014). The SES values are a measure of the deviation of the observed values from those expected by chance; positive and negative SES indicate greater and less dissimilarity between assemblages than expected by chance, respectively. Values greater than 1.96 or less than −1.96 are significantly greater or less than expected, at an α = 0.05. To describe SES trend of each component of β-diversity along the elevation gradient, nonparametric method LOESS with a span = 1.5 were made using the ggplot function of the package ggplot2 (Cleveland & Loader, 1996; Wickham, 2016). To identify the main environmental factors driving changes in species composition along the elevational gradient, we performed the same protocol for species richness, as indicated above, using GLM with beta distribution.

Results

We identified a total of 176 plant species (1,417 = sum of incidence for all species), 32 ant species (485), 48 spider species (206) and 31 bird species (236) collected and/or observed during samplings (see Supplementary Material for species list). The environmental variables showed similar elevational patterns. Mean temperature (DF = 1, Chisq p-value = 5.052e−9), SAVI (DF = 1, Chisq p-value = 1.54e−8), annual mean temperature (DF = 1, Chisq p-value = 2.2e−16) and annual precipitation (DF = 1, Chisq p-value = 2.2e−16) decreased with elevation, while mean water in soil increased with elevation (DF = 1, Chisq p-value = 2.02e−7) (Fig. 1). The correlation was high between most of the environmental variables (Table S1).

Figure 1 Environmental variables along the elevational gradient of Ventana Mountain.

Lines indicate the general linear model (GLM), with a 95% confidence interval. Dots indicate the average values for each elevation band. Temperature = average temperature, main water in soil, SAVI = average SAVI index, mean annual temperature and annual precipitation.

The regression analysis of changes in the species richness of ants, birds and plants along the elevation gradient showed a decreasing trend at high elevations (ants: DF = 2, Chisq p-value = 2.822e−7; birds: DF = 1, Chisq p-value = 1.450e−7; plants: DF = 2, Chisq p-value = 1.312e−6) (Fig. 2). However, ants and plants richness showed a hump-shaped response, with a sharp decrease from the upper half of the elevational gradient onwards. On the other hand, spiders richness did not change significantly along the elevation gradient (DF = 2, Chisq, p-value = 0.378) (Fig. 2). Furthermore, the richness of endemic plants increases with elevation (DF = 1, Chisq p-value = 5.747e−4) (Fig. 2). Sampling coverage was high for ants, birds, plants (greater than 85%), for most altitudinal bands. However, sampling coverage was low for spiders (between 35% and 70%) (Fig. S1). From the multi-model analyses (Table S2), mean temperature was the main driver of species richness for ants (DF = 1, Chisq p-value = 7.023e−6), plants (DF = 1, Chisq p-value = 8.304e−8) and birds (DF = 1, Chisq p-value = 1.441e−5). However, bird richness was also related/associated with SAVI (DF = 1, Chisq p-value = 0.002).

Figure 2 Species richness of the studied taxa (i.e. ants, spiders, birds and plants) along the elevational gradient of Ventana Mountain.

Lines indicate the generalized linear model (GLM) for each taxon, with a 95% confidence interval. Dots indicate the species richness at each sampling site. Points were jittered to avoid overlap.

Species composition showed an increase in total dissimilarity with elevation for ants (DF =1, Chisq p-value = 1.977e−9), spiders (DF = 1, Chisq p-value = 9.001e−4), birds (DF = 1, Chisq p-value = 5.910e−9) and plants (DF = 1 |, Chisq p-value = 1.890e−9) (Fig. 3A). Species turnover increased with elevation in spiders (DF = 1, Chisq p-value = 9.657e−4), birds (DF = 1, Chisq p-value = 0.027) and plants (DF = 1, Chisq p-value = 1.382e−12) (Fig. 3B), but no significant trends were observed in ants (DF = 1, Chisq p-value = 0.554) (Fig. 3B). On the other hand, the nestedness component increased with elevation in ants (DF = 1, Chisq p-value = 8.665e−6), but decreased with elevation in spiders (DF = 1, Chisq p-value = 3.479e−4) and plants (DF = 1, Chisq p-value = 6.424e−9), while it showed no significant trends for birds (DF = 1, Chisq p-value = 0.801)(Fig. 3C). There was a near-identical pattern of results for standardized beta diversity indices for most taxa, except for ants, β-diversity indices for ants showed a weak hump-shaped pattern (Fig. S2). However, most of the results fell within the range expected by the null model of assemblage formation (−1.96 < SES < 1.96). From the multi-model analyses (Table S2), mean temperature was the main driver of species turnover for spiders (DF = 1, Chisq p-value = 2.539e−3) and plants (DF = 1, Chisq p-value = 1.448e−13), and species nestedness for ants (DF = 1, Chisq p-value = 1.594e−6), spiders (DF = 1, Chisq p-value = 2.914e−3) and plants (DF = 1, Chisq p-value = 1.224e−6). Annual mean temperature also explained nestedness in plants (DF = 1, Chisq p-value = 3.553e−3). However, no environmental variable significantly accounted for species turnover in birds, the closest being SAVI (DF = 1, Chisq p-value = 0.05134).

Figure 3 Taxonomic dissimilarity and its components between the lowest altitudinal band and each sampling site of the upper bands.

Lines indicate the generalized linear model (GLM) for each taxon, using beta distribution with a 95% confidence interval. The x-axis indicates the elevation (masl) of each upper band.

Discussion

Energy, primarily through temperature change, was the major driver of species richness and composition for most taxa, prevailing over productivity and water availability. Most environmental variables decreased with elevation. However, in contrast to annual precipitation, soil water content increased with elevation. This may be due to a local effect or to soil properties (i.e., slope, depth, texture, bulk density and organic matter content), which determines the fraction of precipitation stored (Martínez-Fernández, González-Zamora & Almendra-Martín, 2021). The species richness of all taxa decreased at higher elevations and changes in species composition (β-diversity) were taxon-dependent. Thus, turnover was observed in plants, spiders, and birds, and nestedness in ants. Even though upper elevations harboured lower species richness for most taxa, they provided suitable environmental conditions for species, including endemic ones, that were not found at the lowest sites. These results emphasize the key role played by this isolated and ancient mountain system in preserving regional biodiversity.

Temperature was the main environmental variable explaining the decreasing elevational richness pattern for ants, plants, and birds. This is in agreement with several elevational richness studies involving a single taxon in different habitat types and mountain contexts (e.g., McCain & Grytnes, 2010; Marathe et al., 2020). However, the decline in bird richness was also explained by productivity as suggested by Harrower et al. (2017) via indirect energy action. In our study, the absence of maximum productivity at intermediate elevations suggests that the Ventania System is composed of mountains with wet bases (McCain, 2007; McCain, 2009; Tellería, 2020). This may indicate that water availability was not a limiting factor, while energy could have a differential magnitude of effect depending on the taxon, generating different richness patterns. The peak richness at mid-elevation for ants and plants could indicate the response to several environmental variables simultaneously. However, the changes in richness could also be due to stochastic causes as predicted by the mid-domain effect (Nogué, Rull & Vegas-Vilarrúbia, 2013), due to the random overlap of species with different ranges in a limited area (Colwell & Lees, 2000). The lack of pattern observed in spider richness along the elevation gradient could be due to a methodological problem since the sampling coverage was low for this taxon.

The comparison of the species composition between the lowest band and each sampling site of the upper bands indicated that taxonomic dissimilarity increased for all taxa. Although temperature was the main driver of these changes, the main component of β-diversity differed among them. The trends from the comparison with null models of assembly formation provide evidence in favour of species turnover as the most important component for spiders, birds and plants. Species turnover has been reported as the major driver of changes in species composition along elevational gradients (Foord & Dippenaar-Schoeman, 2016; Gebrehiwot et al., 2019). Moreover, this component has been generally associated with changes in habitat type (Foord & Dippenaar-Schoeman, 2016; Aisen et al., 2017; Iijima & Morimoto, 2021). Our results show that species turnover may also occur within the same habitat type along an elevational gradient. This might be explained by the combined effect of the two processes. First, climatic changes during the last millennia could have led to a latitudinal shift between the Patagonian (steppe), Espinal (xerophytic forest) and Pampas (grassland) ecoregions. As a result, species from different ecoregions would have colonized the Ventania System at different historical times, and some of them could have persisted due to the high environmental heterogeneity of these mountains (Frangi & Bottino, 1995). Second, the speciation events that occurred in these ancient mountains most likely account for the presence of endemic species at different elevations (Cuevas & Zalba, 2009). The loss of ant species along the elevation gradient was not corroborated when compared with null models, this could indicate that the observed pattern does not differ from that expected by chance. However, the increase in nestedness of ant assemblages suggests that low temperatures at high elevations would have prevented many ant species present at the base from inhabiting upper elevations. The origin and initial diversification of most neotropical ant lineages occurred during the expansion of tropical forests (Eocene > 30 Ma) (Andersen & Vasconcelos, 2022). Although some lineages more recently diversified into dry habitats, there may be few grassland species adapted to more extreme environmental conditions (Andersen & Vasconcelos, 2022). In conclusion, this process may result in turnover in species composition due to the presence of specialists (Marathe et al., 2021); or in nestedness when only a few base-dwelling generalist species can withstand the environmental conditions at upper elevations.

The taxa present in this ancient and isolated mountain system are expected to be differentially affected under the current global warming scenario (Barros et al., 2015), as temperature and precipitation are expected to increase for the region (Llopart et al., 2014; Barros et al., 2015). In this line of reasoning, ant richness would increase in mountains top, as a response to the relaxation of environmental filters that limit base species to invading upper elevations, while the diversity of plants, spiders, and birds would decrease due to the loss of species only found at upper elevations. The latter alternative could be due not only to the new unfavourable environmental conditions but also to the competitive exclusion of lower-elevation species over upper-elevation species (Freeman, Strimas-Mackey & Miller, 2022). Mountain ecosystems are biodiversity hotspots especially threatened by global change (Löffler et al., 2011; Rahbek et al., 2019), because of the presence of species with small distribution areas and high levels of specialization (Viterbi et al., 2020). Finally, global warming may lead to biota homogenization in the Ventania Mountain System because it may promote a decrease in β-diversity and endemism but, at the same time, an increase in species richness due to the rise of species currently only found at low elevations.

Supplemental Information

Supplemental Information 1 Supplementary figures and tables

Supplemental Information 2 Raw data and scripts

The scripts refer to different topics addressed in the article: Environmental description, environmental variables and diversity (one file), Alpha diversity (two files) and Beta diversity (two files). The arrays with raw data are called automatically by the scripts.

M. Apellaniz, H. Iuri, P. Berge, P. Cambiaggi and P. Pairo for field and lab assistance. Rangers of the Ernesto Tornquist Provincial Park provided logistic support. Three anonymous reviewers provided useful comments that improved the manuscript.

Additional Information and Declarations

Competing Interests

Author Contributions

Field Study Permissions

Data Availability

The authors declare there are no competing interests.

Santiago Santoandré conceived and designed the experiments, performed the experiments, analyzed the data, prepared figures and/or tables, authored or reviewed drafts of the article, and approved the final draft.

Carolina Samanta Ramos performed the experiments, authored or reviewed drafts of the article, and approved the final draft.

Pablo Picca performed the experiments, authored or reviewed drafts of the article, and approved the final draft.

Julieta Filloy conceived and designed the experiments, performed the experiments, authored or reviewed drafts of the article, and approved the final draft.

The following information was supplied relating to field study approvals (i.e., approving body and any reference numbers):

Organismo Provincial para el Desarrollo Sostenible de la Provincia de Buenos Aires.

The following information was supplied regarding data availability:

All the scripts and raw data used in the study are available in the Supplemental File.

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
