# Peer review of "Taxon-dependent diversity response along a temperate elevation gradient covered by grassland"

_PeerJ, doi:10.7717/peerj.17375_

## Round 0.1 · original submission · Major Revisions

Dear Dr. Santoandre,

After the evaluation by three independent reviewers, your manuscript may be accepted for publication in PeerJ after major reviews are performed on the current version of your study.

All three reviewers raised important issues in your study that must be addressed. Please prepare the next version of your manuscript and a rebuttal letter informing the point-by-point changes that were included in your manuscript and those that were disregarded. Please do not forget to provide justifications of why such changes were not considered in the next version of your manuscript.

SIncerely,
Daniel Silva

Reviewer 1 ·

Basic reporting

The paper “Taxon-dependent diversity response along a temperate elevation gradient covered by grassland” aimed to investigate how changes in climatic conditions along an elevational gradient affect the diversity of four different taxa. The paper is relevant and particularly interesting as it explores the energy hypothesis across an elevational gradient where no significant changes in the habitat type occur. It is well-structured and written in unambiguous and technically correct English. Figures and the Table are relevant to the content of the article (although I suggest some changes -see next sections-).

Experimental design

The research question is relevant and meaningful, and the experimental design is correct. However, I have some concerns about the independence of observations within an elevational band. Is 50m enough to take these samples as independent for ants and spiders? (ln 154). Please explain why they may be considered independent for these organisms or whether they should be taken as pseudoreplicates. In addition, more than one pitfall trap per site seems necessary for diversity estimation. Regarding birds, is 150 meters sufficient to consider the observations as independent? (ln 166). The “Handbook of field methods for monitoring landbirds” (Ralph et al. 1993) recommends at least 250m between forest sampling points and more than 250m in open habitats. To avoid pseudoreplication, I suggest calculating species richness (for all taxa) as the number of species of each taxon per sampling band (N=6) instead of per sampling site.

On the other hand, more information is needed to understand the variables used in the analysis fully. Could you please explain (and justify) how you estimated temperature and humidity for each elevational band? (ln 198-201). Did you use a maximum/minimum value or an average? Did you use one sensor per elevational band? How did you select where to place it? These kinds of sensors are susceptible to local conditions. Are your local records correlated with temperature and water availability measures extracted from WorldClim? Please explain why the estimate with sensors would be better than WorldClim's. I suggest including WorldClim variables in the analyses or showing the robustness of sensor measures in an Appendix.

In addition, please justify why you preferred to use single-variable models instead of adjusting models with all possible combinations of variables that would make it possible to evaluate the interaction between variables. Are these variables highly correlated?

In order for the reader to see the differences between the models, I suggest all models be shown in Table 1 (or in an Appendix).

It is hard for me to see the sharp decrease for ants in Fig. 2 (ln 225-226). I suggest remaking Fig. 2, showing one panel for each taxon. Symbols used and differences in species richness (mainly plants versus other taxa) make visualization difficult. In addition, ants and plants data could better adjust to a hump-shaped pattern?

Validity of the findings

For the findings to be considered valid, I think the authors should clarify their experimental design and statistical analyses or reanalyze data based on my previous suggestions.

Additional comments

For a better understanding, the second paragraph needs more detail (it is only two sentences). Please expand the idea.

Ln 48-49 and ln 65-67 express the same idea. Please remove from one paragraph

Ln 53 remove “(“ after “;”

Ln 57-77 I suggest splitting this paragraph in two, one focusing on elevational diversity patterns (ln 57-67) and the other one focusing on habitat types' influence on diversity patterns (ln 67-77). Also, I suggest explaining (in the former) the species richness patterns proposed by McCain (2007) for changes in water availability.

Ln 79-100 For readability, I suggest splitting this paragraph in two: 79-90 and 90-100

Ln 109-111. Please be more specific. Which climatic conditions? How do you relate these climatic conditions with the energy hypothesis?

Ln 132. You say that the sampling covers the entire gradient. However, in line 128, you describe the range from 450 to 1200 masl, but the highest sampling site is at 1001 masl. This implies that you left the top 25% of the gradient out of the sampling, which might affect conclusions about observed patterns for richness and environmental variables. Please clarify this issue and elaborate in the discussion section.

Ln 135. Please specify the sampling year.

Ln 192. Please remove “Then”

Ln 242-249. You emphasize the role of this mountain on endemic biodiversity. Could you elaborate on endemic species in the results section?

Ln 259. Is it correct the word “different” here? Could you explain?

Ln 273. I suggest changing “probably” to “. This might be explained by …”

Ln 292: “ant richness would increase” at mountain top?

Ln 301: “… an increase in species richness” Are you expecting an increase in species richness in the Ventania Mountain System? Please explain

Ln 290-301 You devote a whole paragraph to global warming in the discussion, but this topic does not appear in the introduction. For consistency, I think the issue should be mentioned in the introduction.

·

Basic reporting

The English is clear, although there are a few instances of awkward sentences and wrong words. some of the sentences could improve using better style, but I've highlighted these throughout the manuscript. The references are sufficient, but I would have liked to have seen mention of the other processes that might affect species richness in particular, such as the role of neutral processes, the mid-domain effect, area and mid-point attractors. Overall the manuscript is well structured with relevant tables and figures. No raw data were shared, and the species lists provided can be improved by including abundance for species at each elevation and indicating which are endemic. At the moment, they only list the names of the species. Mention of these endemic species is made in the discussion, but none of the results are explicitly dealt with in the supplementary table. How many endemics, where do they occur along the elevations, and what are the levels of endemicity?

Experimental design

Well-designed sampling protocol, with six transects and a range of sampling techniques. A more detailed description of the transects and sampling protocol would improve the manuscript though, e.g. how far apart are these transects, when (which season was the sampling done? Or was it across several seasons? Most importantly, though, there is no assessment of sample coverage; the number of individuals caught seems to be very low, and for the spiders in particular, this could mean that they were undersampled. Coverage (Jost & Chaoe 2012) provides a ready measure to assess sampling. Many of the predictor variables seem to also co-vary, and even though the temperature had the best model, there is no mention of the AIC values for the next best models or even just the null model - so no context to evaluate the best model reported.
The beta-diversity analysis would have to use null models to assess whether the elevational distance significantly affects the beta-diversity measures modelled - see Bishop, Tom R., et al. "Contrasting species and functional beta diversity in montane ant assemblages." Journal of Biogeography 42.9 (2015): 1776-1786. and Foord & Dippenaar-Schoeman 2016. In particular, because these beta diversity measures are not independent.

Validity of the findings

Although the analysis has to be complimented by the analyses I mentioned above, I don't think the findings would change materially. Some of the results might not be significant, but the general trends should remain.

Additional comments

There are some very interesting findings in this study; these include the contrast between what drives ant beta diversity and that of the other taxa, but this is not a requirement for publication in PeerJ, so it would be important to deal with providing access to the raw data, summarizing the species tables in a meaningful way, a clearer explanation of the sampling design, including coverage into the models (at the moment there is no real estimate of coverage, which is particularly important for the taxa for which sampling resulted in very low numbers of individuals and including null models in the analysis of beta diversity.

Reviewer 3 ·

Basic reporting

The manuscript entitled “Taxon-dependent diversity response along a temperate elevation gradient covered by grassland” is an interesting contribution that shows richness patterns (both alpha and beta) of four taxonomic groups: ants, spiders, birds and vascular plants along elevation gradient of the Ventana Mountain in Argentina. It is a local study, but aiming at testing general hypotheses on biogeographical patterns along an elevation gradients. The strength of the study is the multi-taxon approach and location in the understudied region, in the remnants of natural grasslands, which in other places were heavily transformed by agriculture. The language can be improved, but in general the manuscript is easy to follow. The structure and wording can be improved e.g.
- The aim of the study is poorly formulated as it not just ‘investigation’ (as it is formulated in the text) but evaluation of some hypotheses or answering specific research questions. Taking into account the scope of the study it should be possible to formulate 2-3 research questions or hypotheses instead of the given statement.
- A structure of the Introduction can be slightly improved, as in the current version paragraphs are quite long and cover several aspects each. For example the information about mountains as perfect places for studies on diversity patterns can be gathered in one paragraph instead of being scattered throughout all the Introduction.
- It should be specified at least in the methods section that authors sampled only vascular plants species (to clarify that bryophytes were not studied)
- Keywords: should be written in a singular form (climatic driver instead of drivers)
- Sentence in lines 83-85 should be reformulated
- Line 134: better write range than “approximately 100 m” (as sometimes it is quite far from that number…)

Experimental design

In the current manuscript some key information are missing, what make the proper assessment of the study methods difficult. These points need to be clarified or better elaborated before the manuscript may be concern for publication.
- Many major concerns regards the environmental variables. There is no details on sampling of environmental data: Did the location of the devices measuring temperature and humidity was standardized at different elevation belts? As they were located close to the ground, vegetation structure or exposition could impact the obtained measurements. Why it was measured at only 6 h intervals and only for one month? What was the starting date? It is quite short time and such data can be proper to conclude about weather conditions in that specific month, but it is hard to conclude on their basis on long-term climatic conditions.
- There is no explanation how the average relative humidity is related with water availability, which is the term used throughout the text. The relative humidity is strongly dependent on temperature, so those two measures are not independent variables, as it can be interpreted from the text. Moreover, the relationship between elevation and average relative humidity seems to be very weak (as shown on Fig. 1). What was the parameters and strength of that relationship? If there is a weak relationship between relative humidity and elevation it is hard to expect that species richness at different elevation belts will show strong relationship with relative humidity.
- The only environmental variables analysed in the study are locally measured temperature and average relative humidity, as well as primary productivity obtained from soil-adjusted vegetation index. Small scale richness patterns are usually controlled by multiple, local environmental factors (e.g. Turtureanu, 2014; Polyakova, 2016). In case of plants, their richness within 1 m2 can be very different depending on the exposition, inclination, occurrence of local disturbance or heterogeneity (rock outcrops etc.), soil depth. For example Fig. 2 shows that at the lowest elevation the range of species richness of plants was very wide – from ca. 13 to ca. 45 species, so it indicates very large variations at that elevation (in this case there is also a risk of heteroscedasticity in the regression analysis that should be tested). Similarly, ant and spider species richness can be determined by fine-scale factors and vegetation structure. Did the selection of sampling points allowed to exclude an impact of other variables on species richness of the studied groups? It is particularly important in such a local study, where only one transect represents each elevation belt (thus there is no replications over other areas). If some additional environmental data were collected (as slope inclination, exposition) or can be easily obtained, then I would recommend to include them in the analyses.
- In case of plants and ants the relationship between richness and elevation seems to be unimodal. It is interesting finding suggesting that the temperature is not the only driver of the species richness patterns and it should be discussed.
- What is the resolution of primary productivity data? Probably for analyses at larger spatial scales and for some taxonomic groups like birds, low resolution of such data is not a problem. But, for plants studied in small plots, spiders and ants the quality (resolution) of such data can be inappropriate for analyses of small-scale richness patterns. Please add more details on that aspect.
- In the paragraph “Sampling of ants and spiders…” there is no information whether those groups were sampled in the same locations as birds and plants. With such multi-taxon approach performed in one research area it is important to ensure that sampling was performed in the same locations (to exclude possibility that the results are biased by different environmental conditions at the same elevation).
- An approach in the analyses of species composition dissimilarity is a bit unclear: what was the purpose of pooling the species from the lowest elevation? Why did the dissimilarity was not calculated just between pairs of plots? How the dissimilarity was calculated? There are many ways of calculation dissimilarity so proper references or formulas should be given.

References:

Polyakova, M.A., Dembicz, I., Becker, T., Becker, U., Demina, O. N., Ermakov, N., ... Dengler, J. (2016). Scale-and taxon-dependent patterns of plant diversity in steppes of Khakassia, South Siberia (Russia). Biodiversity and Conservation, 25, 2251–2273.
Turtureanu, P.D., Palpurina, S., Becker, T., Dolnik, C., Ruprecht, E., Sutcliffe, L. M. E., … Dengler, J. (2014). Scale- and taxon-dependent biodiversity patterns of dry grassland vegetation in Transylvania. Agriculture, Ecosystems and Environment, 182, 15–24.

Validity of the findings

Some of the finding can be biased due to lack of control over other environmental variables. Several methodological points were poorly described, what makes the proper assessment difficult - please see the comments above.

---

## Round 0.2 · Major Revisions

Dear Dr. Santoandre,

After the evaluation by one additional reviewer in this second review round, the reviewer believes your manuscript may be accepted for publication in PeerJ after major reviews are performed on the current version of your study. Please prepare the next version of your manuscript and a rebuttal letter informing the point-by-point changes that were included in your manuscript and those that were disregarded. Please do not forget to justify why such changes were not considered in the next version of your manuscript.

Sincerely, Daniel Silva

Reviewer 4 ·

Basic reporting

Please see my general comments

Experimental design

Please see my general comments
Fieldworks!

Validity of the findings

Please see my general comments

Further analyses are needed!

Additional comments

I have reviewed the manuscript entitled "Taxon-dependent diversity response along a temperate elevation gradient covered by grassland" for possible publication in PeerJ.

A notable strength of the paper lies in its comprehensive examination of multiple taxa and assessment of various components of diversity, including alpha and beta diversity across different taxonomic groups, ranging from plants to ants, birds, and spiders. This breadth of taxonomic coverage enriches the study's depth and contributes to a more holistic understanding of diversity patterns along the temperate elevation gradient covered by grasslands. Another commendable aspect of the paper is its focus on an underrepresented area. I have carefully reviewed all comments raised by three reviewers and the provided answers by the authors. The authors attempted to improve the manuscript, but some shortcomings still exist and should be addressed in the next revision.

Abstract:
- While the paper demonstrates strength in its diverse taxonomic coverage, the abstract could be refined for clarity and relevance (Lines 15-17). The mention of different habitats may be misleading since the study primarily focuses on grassland habitats within the mountain system. Aligning the abstract more closely with the actual scope of the study would enhance its accuracy and effectiveness in conveying key findings to readers.

Additionally, the first sentence of the abstract indicates that the authors worked on animal taxa, but they actually examined a range of taxa from animals to plants. Therefore, this sentence should be improved.

- The last sentence of the abstract (Lines 27-30) is not applicable to the content of the paper. The paper focused on species richness and diversity patterns along elevation but did not study climate change and its potential effects on the future distribution of various taxa. This information might be suitable as part of the conclusion and suggestions section, but it is not a part of the current results.

Introduction:
In Lines 43-46, the focus on animals is highlighted despite the study's exclusive focus on grasslands and not other habitat types. The structure of the introduction should be reformulated and restructured to reflect this.

In Lines 59-61, the sentence implies that your mountain systems have a moist and warm basis, correct?

In Lines 95-98, the statement regarding different models predicting an increase in temperature and annual precipitation for the region suggests an increase in precipitation due to climate change?

Study Area:
The description of the study area should be better organized. The sentence (Lines 123-125) should be placed at the beginning of the study area section to provide a general overview before delving into specifics such as protection status, geographical limits, biodiversity, geology, or climate.

Methods and Results:
Clarifying the definition of endemic taxa is necessary to ensure reader comprehension and avoid ambiguity (Line 231). Providing explicit criteria for what constitutes endemic taxa, whether restricted to the mountain system or a specific region, would enhance the clarity and accuracy of the study's findings.

I am skeptical about how PCA helped in the analysis. With only six averaged values for some environmental variables and high correlations between variables, a thorough examination of PCA's contribution to understanding the study outcomes would strengthen the methodological approach and data interpretation. Further elucidating the interpretation of PCA results and considering alternative analytical approaches, such as regressions and GLM analyses, could provide deeper insights into the study's findings and enhance their ecological relevance.

Addressing the negative correlation between water availability and precipitation is crucial for preventing potential misinterpretations. The higher elevation sites have more water availability but less precipitation. This might make the statement in the first sentences of the discussion, suggesting that increased water availability could affect species richness, misleading and ambiguous.

My general comment:
The interpretation of the paper seems quite arbitrary without a detailed examination of the main questions regarding underlying ecological factors. Climate factors might overshadow other factors, such as soil factors. While focusing solely on climatic factors is acceptable, it becomes problematic when dealing with highly correlated factors. My suggestion is to eliminate some highly correlated variables and focus on relevant and ecologically interpretable ones, then interpret and justify the results accordingly. Currently, the PCA results seem superfluous and obvious. Perhaps implementing regressions and GLM analyses on the trend of richness and beta diversity and their associated ecological factors for each group of taxa separately could provide more insights. Additionally, considering other ecological factors like land use/management and soil, and explaining how the sampling strategy addresses these factors would enhance the paper's quality. Although I understand the limitations regarding expenses, the sampling methodology should be explained more explicitly to ensure consistency across elevation bands in terms of soil and land use.

---

## Round 0.3 · accepted · Accept

Dear Dr. Santoandre,

I am pleased to inform you that your manuscript has been accepted for publication in PeerJ.

Congratulations!

Reviewer 4 ·

Basic reporting

No comments. Revised materials are satisfying

Experimental design

No comments. Revised materials are satisfying

Validity of the findings

No comments. Revised materials are satisfying

Additional comments

No comments. Revised materials are satisfying